# Improved Efficiency Based on Learned Saccade and Continuous Scene Reconstruction from Foveated Visual Sampling

**Jiayang Liu[1], Yiming Bu[1], Daniel Ts'o[2], Qinru Qiu[1]**
[1]Department of Electrical Engineering and Computer Science, Syracuse University
[2] SUNY Upstate Medical University
{jliu206, ybu104, qiqiu}@syr.edu, {tsod}@upstate.edu

## Abstract

High accuracy, low latency and high energy efficiency represent a set of conflicting goals when searching for system solutions for image classification and detection. While high-quality images naturally result in more precise detection and classification, they also result in a heavier computational workload for imaging and processing, reduced camera frame rates, and increased data communication between the camera and processor. Taking inspiration from the foveal-peripheral sampling mechanism, and saccade mechanism of the human visual system and the filling-in phenomena of brain, we have developed an active scene reconstruction architecture based on multiple foveal views. This model stitches together information from a sequence of foveal-peripheral views, which are sampled from multiple glances. Assisted by a reinforcement learning-based saccade mechanism, our model reduces the required input pixels by over 90% per frame while maintaining the same level of performance in image recognition as with the original images. We evaluated the effectiveness of our model using the GTSRB dataset and the ImageNet dataset. Using an equal number of input pixels, our model demonstrates a 5% higher image recognition accuracy compared to state-of-the-art foveal-peripheral based vision systems. Furthermore, we demonstrate that our foveal sampling/saccadic scene reconstruction model exhibits significantly lower complexity and higher data efficiency during the training phase compared to existing approaches. Code is available at Github.

## 1 Introduction

High resolution, low latency and high energy efficiency represents a set of conflicting requirements for cameras in an image sensing and processing system. While utilizing high resolution images usually leads to more precise detection and classification, it reduces frame rate, extends the read out time and increases the amount of data communications of the camera. Biological visual systems outperform artificial optical sensing and processing systems in precision, responsiveness, and energy efficiency. This partly comes from the fovea of the retina, the saccadic eye movements that permit high resolution sampling of a visual scene, and how the brain utilizes the information obtained from foveal and peripheral region.

The visual field of the human eye spans approximately 140 degrees of arc, where only 1 degree in its central are observed in high resolution called fovea, and the rest of the visual field is sampled at diminishing resolution out to the periphery (Bull, 2014). Compared to today's camera images, this foveal/peripheral visual architecture has high sparsity and requires much less input data. It has also been demonstrated to have superior adversarial and occlusion robustness(Deza & Konkle, 2020; Gant et al., 2021; Vuyyuru et al., 2020; Harrington & Deza, 2022; Wang et al., 2021). Sparse input can effectively reduce computation and memory requirements, however, it also accompanies with significant accuracy loss (Lukanov et al., 2021; Fang et al., 2020; Zhang et al., 2023; Kaplanyan et al., 2019). The accuracy degradation can partly be mitigated by having a high resolution foveal view and a low resolution peripheral view. The distribution of information carried by differing portions of an image is highly nonuniform and regionally imbalanced. Placing the foveal center at a

high information region and the peripheral vision in the low information region will result in better perception without significant increase in the amount of input data and computation.

Although humans receive visual information with drastically reduced resolution in the periphery, we seem to see sharply and clearly. This is because our eyes frequently and rapidly move the foveal centers to place them on different regions, and our brain stitches them together and fills in the rest in a gross, impressionistic way based upon what we know and expect (Ehinger et al., 2017). When humans observe their environment with purpose, like during object recognition or visual search, prioritizing a saccade towards an informative fixation further enhances both the speed and accuracy of scene understanding (Shepherd et al., 1986; Moore & Fallah, 2001)(Johnson, 2021). The brain's filling-in capability allows us to complete missing information across the physiological blind spot, and across natural and artificial scotomata, yielding visual perception that is continuous and seamless. The sensing behavior and the information collected are closely coupled to each other, together delivering incomparable performance and energy efficiency.

Inspired by the foveation and saccade mechanisms of the human visual system, and the filling-in phenomena of brain, we present an artificial vision system designed for energy-efficient and low cost sensing and processing. The system employs a foveal/peripheral vision-inspired image sampling incorporating saccadic control to reduce the amount of data required from the camera. The incoming stream of foveal and peripheral inputs is processed for scene prediction and reconstruction where the missing pixels are filled-in to form a smooth and semantically consistent image. The scene reconstruction model is trained using self-supervised learning to maximize the structural similarity and minimize the mean square error between the constructed and original images. Based on the received information, the saccade controller chooses the next foveation target such that the scene recognition can be completed with the minimum amount of input data and the highest accuracy. While there are many forms of scene recognition, without loss of generality, we focus on image classification. Our experimental results show that the scene reconstruction improves the accuracy of image classification by 64.8% on GTSRB dataset and 35.2% on ImageNet dataset. And the controlled saccades provide an additional 2.9% and 11.1% improvement for GTSRB and ImageNet respectively. Overall we can reach a similar image classification results at 70% less pixel usage.

To the best of our knowledge, this is the first work that merges multiple foveal/peripheral based vision samples controlled by saccades into a reconstructed image. While some of the existing works adopts foveal/peripheral vision-inspired image sampling (Lukanov et al., 2021; He et al., 2021; Uzkent & Ermon, 2020; Cheung et al., 2016), they do not consider sequential saccades and neither do they perform scene reconstruction. Instead, new classifiers are trained directly on the pixels received from foveal/peripheral regions, which requires a significantly amount of labeled data and tuning efforts. Compared to those works, an important advantage of our approach is high data efficiency during the training phase. The self-supervised training of scene reconstruction model does not require any labeled data. The training of the RL based saccade model can be done without the labels if the goal is to maximize the similarity of the reconstructed and the original images. The training process does require labeled data if the goal is to maximize the classification accuracy. However, we also found that the saccade model is highly transferable. In other words, the pattern of saccades moving the foveal center is similar under different scenes. Using the training data that covers only 50% of classes in the ImageNet dataset, we can train the saccade model that works equally well as the model trained using all training data. Overall, our model and sensing mechanism can easily be used to replace the front-end of existing artificial visual systems without the necessity of modifying the back-end image classification or object detection model. The contribution of this work can be summarized as follows:

- This is the first framework that integrates foveal-peripheral sampling and saccade control with continuous scene reconstruction. Experimental results show that the proposed sampling and pre-processing framework achieves similar image recognition performance with at least 70% fewer pixels compared to systems with conventional imaging front-end.

- The self-trained scene reconstruction model restores the original scene from highly sparse input and achieves an average of 0.88 structure similarity on ImageNet dataset. This process restores missing information to the sparse input and improves the classification accuracy by at least 35.2%.

- Trained using actor-critic reinforcement learning, our saccade controller additionally improves the scene classification accuracy by more than 10% . And it is highly transferable. It works effectively on scenes that are different from the training classes.

- Compared to the state-of-the-art foveal-peripheral based vision systems, with the similar amount of input pixels, our framework gives 5% higher scene classification accuracy while requires 50% less training data.

In the rest of the paper, we use the term *foveal-peripheral view* to refer to the pixels sampled from the foveal and peripheral region. We use *glance* and *glimpse* to refer to the action of taking a sample. We use the term *foveal-peripheral sampling* to refer to the sampling strategy (i.e., high density sampling in the foveal region and low density sampling in the peripheral region.)

## 2 BACKGROUND AND RELATED WORK

### 2.1 FOVEAL-PERIPHERAL INPUT IN HUMAN VISION SYSTEM

Human eyes typically capture much more information than our brains are capable of processing (Borji et al., 2019). To improve the efficiency of information transmission and processing, the human visual system has develop multiple mechanisms to ensure the eyes move to where they are most needed. There are primarily two types of photosensitive cells in the eyes responsible for retinal image capture. They are distributed non-uniformly in the retina, with the cones providing color and high acuity central vision, while the rods specializing in low-light vision (Roorda & Williams, 1999). There are about 6 million cones and 125 million rods in each human eye. More remarkably, this retinal information is highly processed and compressed, culminating to only 1.2 million separate outputs (axons) in each optic nerve. While we possess high acuity in only a very small portion of our entire visual field, we rarely suffer from any obvious deficiency due to our foveal and saccadic mechanisms(Hirsch & Curcio, 1989).

The fovea centralis is a small pit structure in the eye, composed of closely packed cones. People can see objects clearly only in this small region which is 1-2 degrees in the field of vision. Outside the fovea centralis, the spatial resolution of human vision drops dramatically. To continuously collect useful information, our neural system processes the visual information at the current fixation site and moves the foveal center to new locations where further information might be found. Such process is referred to as a saccade. The correlation between foveation and saccade has been extensively studied and confirmed by many studies (Poletti et al., 2013; Shepherd et al., 1986; Moore & Fallah, 2001; Henderson & Ferreira, 1990). Biological studies prove that saccade planning is related to attentional signals generated by the brain (Shepherd et al., 1986; Moore & Fallah, 2001) based on the information provided in periphery. Experiments also indicates that saccade planning helps the brain process visual stimuli more efficiently.

### 2.2 MACHINE LEARNING FOR ARTIFICIAL FOVEATION AND SACCADE

Many artificial visual systems learned from the biological foveation and saccade processes to reduce data processing and transmission overhead and to enhance image understanding (Zhao et al., 2018; Elsayed et al., 2019; Uzkent & Ermon, 2020; Oord et al., 2018; Deza & Konkle, 2020; Wang & Cottrell, 2017). For example, (Jaramillo-Avila & Anderson, 2019) showed that using the foveated sampling, the image size can be reduced to 1/16 of its original value which doubles the frame rate of object detection. At the same time the percentage recall of object detection in the foveal center drops from 34% to 24% while the percentage precision increases from 25% to 30%. Overall, the performance of object detection on foveated image is much better than that on a uniformed subsampled image. However, the recall and precision of object detection out in periphery drops rapidly to only 10% of the baseline. This work simply imposes a fovea centralis to the center of the image without saccades. The authors of (He et al., 2021) subsample an image by taking some random patches, which can be viewed as a set of foveal views. They then reconstruct the image using a masked autoencoder trained to minimize the MSE error of the reconstruction. Although the autoencoder is trained via self-supervised learning, they do need to extensively train and fine tune the classifier. This work does not consider how to sample the patches for better performance.

Determining the best location of the foveal center is similar to locating an optimal region for attention. A simple way is to move the foveal center to the regions with the most salient image features(Wloka et al., 2018). However, such a saccade model ignores any top-down influences of attention. Lukanov et al. (2021) mitigates this limitation by averaging the bottom-up saliency with top-down saliency from the activation map of a classifier. However, at the beginning of the visual process, the classification is likely to be wrong, which will lead to incorrect saccade movements. Schwinn et al. (2022) trained a model to predict the visual scanpath by minimizing the error of the guide application. They assumed foveation-blur model for the input, which is not sparse. (Elsayed et al., 2019) considers a foveal center as a region with hard attention. It applies labeled supervised training to an attention model and moves the foveal center to the region with the highest attention. However, no peripheral vision is included in this work and the attention map is generated solely based on the foveal view. As the result, it is not able to discover interesting regions outside the foveal center and it usually take quite a few number of glimpses in order to gather enough information. (Chen, 2021) learns the strategy of saccades using reinforcement learning. In each decision step, it selects a quadrant of the current input image, refine its resolution, and set the selected quadrant as current image. This process continues until it reaches the desired resolution of the foveal center. The drawback of this approach is the latency. It takes several decision steps to find the location of the foveal center and, for each image, only one foveal center is identified.

For all of the aforementioned works, a new backend model must be trained on the received foveal-peripheral views for classification or object detection. To reach state-of-the-art performance, a large number of labeled data is required. In this work, we present a framework that mimics the foveal-peripheral vision and saccade process. It actively reconstruct the original image from a sequence of foveal and peripheral views and at the same time decides the location of next foveation target. The reconstructed image resembles with high similarity the original image, hence no new back-end model needs to be trained. Any existing classification or object detection model can continue to be used as-is.

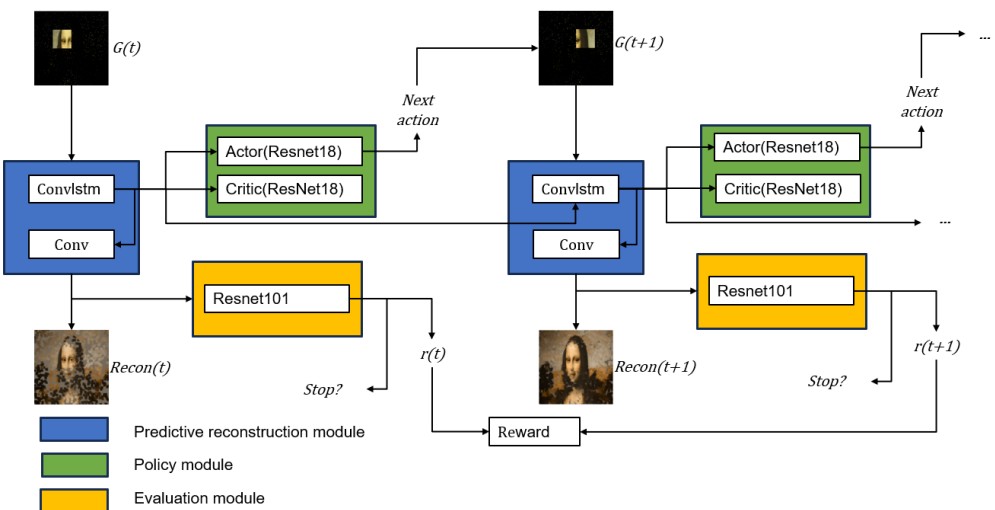

Figure 1: Architecture Overview

## 3 METHOD

The overall architecture of the proposed artificial vision system is comprised of three core components: the predictive reconstruction module, the evaluation module, and the policy module. Figure 1 shows the data flow between the system components. The predictive reconstruction module receives a sequence of foveal-peripheral views. Its primary function is to merge the received information to generate a high-resolution rendition of the original scene. The policy module controls the saccades. It leverages information obtained by the predictive reconstruction module to strategically direct foveal center to a new position. After that, a new foveal-peripheral view is sampled from the environment. This entire process recurs until the termination condition is fulfilled.

## 3.1 FOVEAL-PERIPHERAL VISION

We apply foveal-peripheral sampling to reduce the amount of input for a lower transmission and processing cost. We divide the image into $N \times N$ regions of equal size. For each sampling process, one of them will be selected as the foveal center and the rest will serve as the peripheral region. The pixels in the foveal center will be sampled with probability 1 and the pixels in the peripheral region will be sampled with a very small probably $\mu$. A binary mask $M \in \{0,1\}^{X \times Y}$ will be generated based on the sampling probability to select the pixels, where X and Y are the dimension of the original image. The mask is generated independently for every frame, so that the system can accumulate more context over time. An example of a sequence of foveal-peripheral views can be seen in Figure 2 (c).

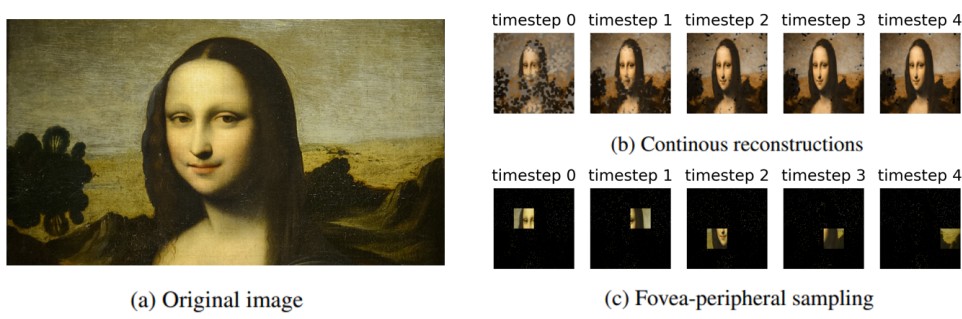

Figure 2: An example of model input(c), model output(b), and original image(a)

## 3.2 PREDICTIVE RECONSTRUCTION MODEL

We adopt a three-layer ConvLSTM as the structure of the predictive reconstruction model, for its superb ability in memorizing spatial-temporal patterns. This model is trained independently to the other two models in a self-supervised manner. Given the foveal-peripheral sampled input as described in section 3.1, the model is trained to reconstruct the original image. During training, we randomly select the foveal center to prevent any bias in the training so that the trained model can work with any saccade controller. More details about the model and data flow are discussed in appendix A.1.

Previous studies (Lotter et al., 2016; He et al., 2021) that aim to reconstruct the original image from a sub-sampled image seek to minimize the mean squared error (MSE) of the reconstruction, as it is simple and inexpensive to compute. However, MSE error is computed on local differences. When we calculate MSE for a pixel at coordinates (x,y), it only considers the differences between the original and predicted image at location (x,y). Given the substantial sparsity in the input data (only 8% of pixels per frame), the limitations of using MSE loss for image reconstruction is obvious. It does not favor predictions with better spatial coherence and higher smoothness. Our experimental results show that the model trained solely on the MSE of reconstructed image yields unsatisfactory results.

We have chosen to use instead the Local Structural Similarity Index(SSIM)(Wang et al., 2004; 2003), which takes into account the structural information and luminance of an image, and aligns better with human perception of image quality. It measures the similarity based on patches taken from the same location of two images being compared. For selected patches x and y, it compares: the similarity of the local patch luminance (brightness values), the similarity of the local patch contrast, and the similarity of the local patch structure.

$$SSIM(x,y) = \frac{(2\mu_x\mu_y + C_1)(2\sigma_{xy} + C_2)}{(\mu_x^2 + \mu_y^2 + C_1)(\sigma_x^2 + \sigma_y^2 + C_2)} \tag{1}$$

Where $\sigma_{xy} = \frac{1}{N-1}\sum_{i=1}^{N}(x_i - \mu_x)(y_i - \mu_y)$, $\mu$ and $\sigma$ are mean and variance of patches to be compared. In this work, we trained the predictive reconstruction model to minimize a hybrid loss of MSE and (1-SSIM). We found that the hybrid loss gives a superior result than using either MSE or

SSIM alone.

$$Loss_{hybrid}(x,y) = MSE(x,y) + \lambda[1 - SSIM(x,y)] \tag{2}$$

Figure 2 (b) gives an example of the sequence of reconstructed image from the corresponding foveal-peripheral views.

### 3.3 EVALUATION MODULE

The evaluation module assesses the performance of scene reconstruction. Its output is used to generate reward for the training of the saccade controller. Different evaluation strategies can be applied. For example, SSIM or MSE can be used as evaluation functions to measure the quality of the reconstruction. The evaluation module could also be an existing image classifier, and the accuracy of the classification can be used to measure the quality of the reconstruction. In either cases, the evaluation module does not require additional training.

### 3.4 SACCADE CONTROL WITH ADVANTAGE ACTOR-CRITIC MODEL

We formulate our problem as a multi-step episodic Markov Decision Process (MDP). The original image is equally divided into several non-overlapping patches, and those patches form the action space of saccade control $x_{ob} = \{x_0, x_1, ..., x_n\}$. The selected foveal center plus the sparse peripheral vision form foveal-peripheral view of one glimpse. The controller observes the environment through the the reconstruction model. In each decision step, the hidden state, $s_{1:t}$, of the reconstruction model, which integrates the present and historical foveal-peripheral views, is presented to the saccade controller to choose the next action, $a_t \in \{x_{ob}\}$. We define the saccad control policy model parameterized by $\theta_p$, as:

$$\pi(a_t|s_{1:t}; \theta_p) = p(a_t|s_{1:t}; \theta_p) \tag{3}$$

Where $\pi(s_{1:t}; \theta_p)$ is a function that maps the observations (i.e., $s_{1:t}$) to a probability distribution over the patch sampling actions $a_t$.

The saccade controller is implemented and trained as an advantage actor critic (A2C) model. Both the actor and critic networks adopt the Resnet18 architecture. The saccade controller is trained after the predictive reconstruction model. During the training of the saccade controller, the predictive reconstruction model remains frozen. The training follows the policy gradient method:

$$\bigtriangledown_{\theta_p} J(\theta_p) = E[\sum_{t=1}^{T} \bigtriangledown_{\theta_p} log\pi\theta_p(a_t|s_{1:t})A(s_{1:t}, a_t)] \tag{4}$$

where $A(s_{1:t}, a_t)$ is the advantage function, calculated as the following:

$$A(s_{1:t}, a_t) = r_t + \gamma V(t+1) - V(t) \tag{5}$$

The reward $r_t$ is generated by the evaluation module. If SSIM is used as the quality metric then it can directly be used as the reward. If the classification result is used as the quality metric, then we set $r_t$ to be the top-1 softmax classification probability if the classification returns correct label, otherwise $r_t$ is set to be the negated top-1 softmax classification probability as a penalty. $V(t)$ is the predicted value from critic model given the state vector $s_{1:t}$, and $V(t+1)$ represents the predicted value after agent take the action $a_t$. The reason that we train the saccade controller using the gradient of the advantage function instead of the absolute reward is because the expected reward of sequence of random sampled actions may have a very large variance, which will lead to unstable training. Using advantage function $A(s_{1:t}, a_t)$ can help to reduce the variance.

To collect more information in a limited number of glimpses, we define the actions of moving the foveal center to a location that has been selected before as invalid actions. This is achieved by using an invalid action mask, $invalidMask \in R^n$. The $i$th entry of the vector is $M$, where $0 < M \ll 1$, if action $x_i$ is an invalid action, otherwise, it is 1. We adjust the action probability using the invalid mask as the following:

$$p'(a_t|s_{1:t}; \theta_p) = Softmax[p(a_t|s_{1:t}; \theta_p) + log(invalidMask)] \tag{6}$$

Equation 6 is differentiable. Hence it can be included as part of the controller without affect the policy gradient flow as shown in Equation 4. For each training and testing image, the initial location of the foveal center is randomly selected.

## 4    EXPERIMENTS

### 4.1    EXPERIMENT SETUPS

We apply the proposed framework on GTSRB and ImageNet(ILSVRC 2012) datasets. To evaluate its performance, we measure the SSIM and classification accuracy of the reconstructed image. We also report the amount of pixels sampled from the camera in order to achieve the accuracy.

The GTSRB (German Traffic Sign Recognition Benchmark) dataset consists of 43 classes of images. We preprocess the images by resizing to a dimension of 112×112 without applying any normalization. We refer these resized images as our original input. Each image is divided equally into 4×4 patches, where each patch serves as a potential region of 28×28 foveal vision. The peripheral sampling portion varies from 1% to 2%. We use a classifier trained with the original input as the evaluation model. It achieves 92.1% top-1 accuracy on the original input.

The ImageNet(ILSVRC 2012) dataset consists of 1000 classes. The original input has size 224×224. Two different sizes of foveal regions are tested. The larger foveal region has the size 56×56, which is obtained by dividing the image into 4×4 patches. The smaller foveal region has the size 32×32, which is obtained by dividing the image into 7×7 patches. A Resnet101 trained using the original input is used as the evaluation model. The model has 77.2% top-1 accuracy on original inputs.

### 4.2    IMPACT OF FOVEAL-PERIPHERAL SAMPLING AND SCENE RECONSTRUCTION

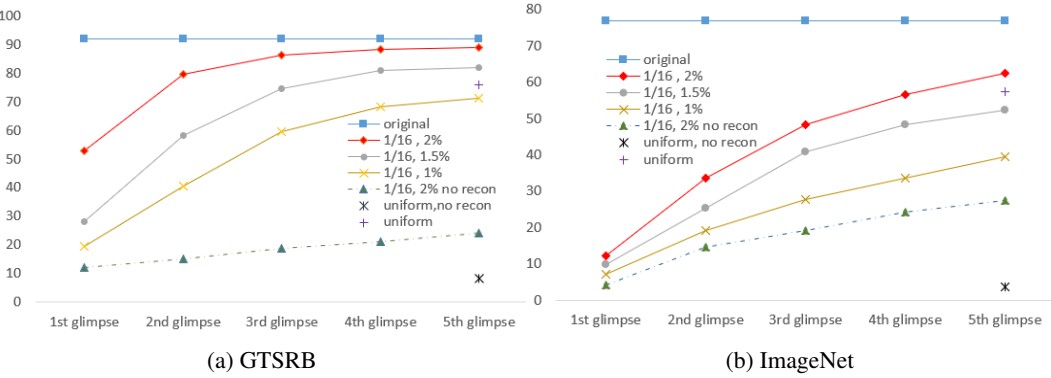

(a) GTSRB                                                    (b) ImageNet

Figure 3: Comparisons of classification accuracy without trained saccades. x-axis is number of glimpse, y-axis is the classification accuracy. All datapoints are classification results of reconstructed images unless specified as "no recon". The data points labeled "uniform" are obtained from uniformly sampled images without any foveal center. The pixel usage of those images are set to be 8.25% per glimpse.

In the first set of experiments, we evaluate how foveal-peripheral sampling and scene reconstruction can improve scene understanding. Random saccades are used in this experiment. For both datasets, we set the foveal region to be 1/16 of the original image and vary the peripheral sampling proportion from 1% to 2%. The foveal centers are randomly picked. After each glimpse, the reconstruction model merges the sampled foveal-peripheral view with the previous input to generate a reconstructed image, which will be classified using the evaluation model. The classification accuracy is reported in Figure 3 using red, grey and orange data points. The blue line in the figure is classification accuracy of the original image, which is also the upper bound of classification.

To demonstrate the impact of scene reconstruction, we directly combine all foveal-peripheral views together without reconstruction and apply the classification model on the combined image, and the result is shown in Figure 3 represented by green triangle data points. As we can see, for both GTSRB and ImageNet dataset, the predictive reconstruction provides a significant improvement in terms of top-1 classification accuracy. For 2% peripheral sampling proportion, after scene reconstruction, the object recognition accuracy on the ImageNet dataset increases from 4.1% to 12.1% at the first glance. At the fifth glimpse, this number is further improved from 27.4% to 62.6%. The fact that the accuracy improvement resulted from scene reconstruction gradually increases as more glimpses

are cast indicates that the benefit of predictive reconstruction accumulates. Similar level of accuracy improvements are observed in the GTSRB dataset.

To demonstrate the efficiency of foveal-peripheral sampling, we compared it with uniform sampling. We uniformly sample 8.25% pixels per glimpse such that it has approximately the same number of pixels as the foveal-peripheral sampling where the fovea size is 1/16 of the original scene and the peripheral sampling proportion is 2%. The classification results on the uniformly sampled images after 5 glimpses are represented in Figure 3 using black data points. Without scene reconstruction, we can see that the foveal-peripheral sampled vision achieves 24.0% and 16.1% higher classification accuracy than the uniformly sampled image for ImageNet and GTSRB datasets respectively. After scene reconstruction, the improvement reduces to 5.6% and 13.6% for ImageNet and GTSRB. The reason that the benefit of foveal-peripheral sampling drops for ImageNet is because the images in this dataset have more complicated scenes. Randomly selected foveal centers may miss some important regions compared to uniformly sampled input.

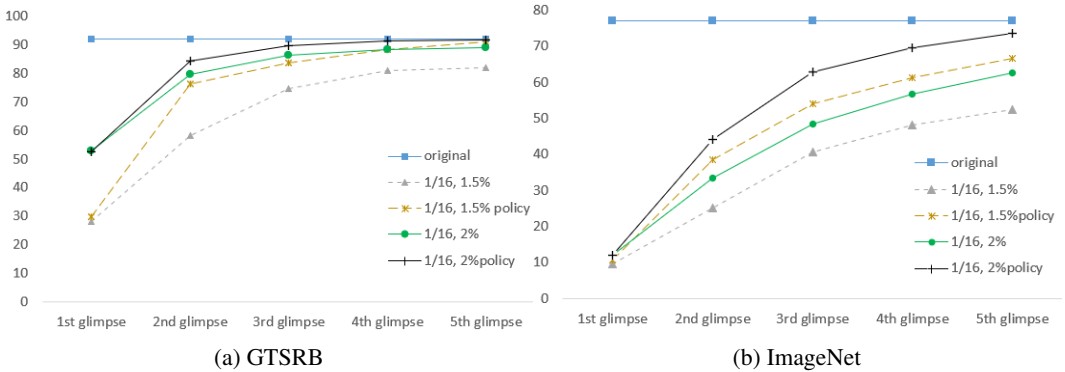

(a) GTSRB        (b) ImageNet

Figure 4: Classification result W or W/O saccade control. The datapoint labeled as "policy" represents the results using the learned saccade control. Size of foveal region is 1/16 of the original image and the peripheral sampling proportion varies from 1.5% to 2%.

### 4.3 IMPACT OF SACCADE CONTROL

In the second set of experiments, we compare the performance of learned saccade controller and a random saccade controller. As shown in Figure 4, with the same number of glimpses, random saccades give lower accuracy compared to learned saccades on both GTSRB (Figure 4a) and ImageNet(Figure 4b). With learned saccades, the classification accuracy for GTSRB achieved 91.1% and 91.8% when we sample 1.5% and 2% pixels for peripheral view respectively. Compared to the results with random saccades, using learned saccades increase the accuracy by 8.6% and 1.9% when the peripheral sampling proportions are 1.5% and 2% respectively. For ImageNet, compared with random saccades, learned saccades improve the top-1 object recognition accuracy by 14.4% and 11.1% respectively when the peripheral sampling proportions are 1.5% and 2%. The results show that the improvement introduced by the learned saccades increases when the peripheral view gets sparser. This agrees with our intuition: when the information from the peripheral view reduces, the location of foveal center becomes more important.

| Model | Test accuracy[%] | Pixel usage rate[%] | Average glimpses |
|---|---|---|---|
| Bio-FCG | 65.17 | 27 | 2 |
| Saccader | 70.0 | 70 | 7 |
| **Ours 7×7** | **70.9** | **28** | **7** |
| **Ours 4×4** | **73.7** | **41** | **5** |
| **Ours7×7early stop** | **66.4** | **22** | **5.4** |
| **Ours4×4early stop** | **69.4** | **30** | **3.5** |

Table 1: Comparing our results with two STOA foveal-peripheral vision system. 7×7, and 4×4 indicate the action space of fovea. The peripheral sampling proportion is 2%.

We further compare our model with state-of-the-art foveal-peripheral visual systems Bio-FCG(Lukanov et al., 2021) and Saccader(Elsayed et al., 2019). In this experiment, we also introduce an early-stop mechanism aiming at a further reduced pixel usage. If the evaluation model confidently predicts the same label in two consecutive glimpses, the process will stop. In the experiment, the confidence threshold is set to be 0.5. As shown in Table 1, our 7×7 model achieved a 5% improvement in accuracy compared to Bio-FCG with similar pixel usage. The Saccader samples at least 70% of the pixels in the original image over 7 glimpse to attain a 70% accuracy, whereas our 7×7 model can achieve slightly better accuracy using only 28% of the pixels in the original image. Moreover, bolstered by the early-stop mechanism, our 4×4 model achieves 69.4% accuracy with only an average of 3.5 glimpses. We need to point out that both Bio-FCG and Saccader have re-trained or fine-tuned their classifier using the foveal-peripheral view, while our framework did not. Re-train the classifier obviously will boost the classification accuracy, however, it also requires significant amount of training data and effort. While our solution allows the users to keep their existing back-end classifier or image processing model as-is.

## 4.4 DATA EFFICIENT POLICY TRAINING

In the third experiment, we compare the performance of the saccade controller trained in different ways. The results are shown in Table 2. It includes a random controller, a controller trained to maximize the structural similarities between the reconstructed and the original images, and a set of controllers that are trained using only a small portion of the training set of the ImageNet. More specifically, we randomly select 100, 200, and up to 500 classes of training data in the ImageNet to train the saccade controller and test the resulting model on the testing data of both the trained classes and all 1000 classes. Finally, we use all 1000 classes of training data to train the saccade controller.

We observed that all of the RL trained controllers outperform the random controller. Secondly, the controller trained to maximize the SSIM does gives the highest SSIM value, however, slightly lower classification accuracy than other RL trained controllers. Thirdly, training the controller using data from only half of the classes in the training set performs almost equally well as training the model using the entire training set. The result shows that the saccade strategy learned from one type of images can be applied to other types of images. Overall, our approach has high data efficiency during the training phase.

| training classes | acc on all classes[%] | acc on training classes[%] | SSIM |
|---|---|---|---|
| Random | 62.6 | - | 0.85 |
| SSIM | 66.3 | - | 0.90 |
| 100 | 70.3 | 77.2 | 0.87 |
| 200 | 71.5 | 75.3 | 0.87 |
| 300 | 72.4 | 74.7 | 0.87 |
| 400 | 73.1 | 74.2 | 0.88 |
| 500 | 73.6 | 73.8 | 0.88 |
| 1000 | 73.7 | - | 0.88 |

Table 2: Transferable data efficient learning result. Random is the top1 accuracy for reconstruction from random multiple saccades.

## 5 CONCLUSIONS AND FUTURE DIRECTIONS

In this paper, we present a novel framework that merges multiple foveal-peripheral views controlled by saccades into a reconstructed image. Our work is an easy-to-use and energy-saving front-end sensing system. It can easily be integrated with any existing back-end processing models, such as image classification and object detection. It reduces 70% of pixel usage to achieve a similar classification accuracy as the original image and requires no effort to re-train or fine-tune the back-end classification model. It also has a 5% more top-1 accuracy compared to state-of-the-art foveal-peripheral based vision systems. Our saccade model is highly transferable and data-efficient. It works effectively on scenes that differ from the training classes. Our future efforts will focus on better periphery sampling techniques and investigating the differences between the trained saccade model versus human visual scanpaths(Kümmerer & Bethge, 2021; Kümmerer et al., 2022)

AKNOWLEDGEMENT

This work is partially supported by the National Science Foundation I/UCRC ASIC (Alternative Sustainable and Intelligent Computing) Center (CNS-1822165).

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

# A APPENDIX

## A.1 IMPLEMENTATION DETAILS FOR PREDICTIVE RECONSTRUCTION MODEL

The predictive reconstruction model has three Convolutional-LSTM layers and one convolutional layer. As shown in figure 1, a foveal-peripheral view ,$G_t$,is received and processed by the predictive reconstruction model in each time step $t$ based on the following equations:

$$h^1_{1:t} = Convlstm^1(G_t, h^1_{1:t-1}) \tag{7}$$
$$h^2_{1:t} = Convlstm^2(h^1_{1:t}, h^2_{1:t-1}) \tag{8}$$
$$s_{1:t} = Convlstm^3(h^2_{1:t}, s_{1:t-1}) \tag{9}$$
$$Recon_t = Conv(s_{1:t}) \tag{10}$$

The variables $h^1_{1:t}$, $h^2_{1:t}$, and $s_{1:t}$ are the hidden states of the ConvLSTM layers, while $Recon_t$ is the predictive reconstruction outcome at time t. In our model, the initial two layers of the ConvLSTM gradually increase the depth of feature maps from 3, to 8 and 16, while the last layer compresses all the generated feature maps into channel dimension 3.

## A.2 HYBRID RECONSTRUCTION LOSS AND NUMBER OF GLIMPSES

When more glimpses are obtained, the SSIM of the reconstructed image increases, while MSE and hybrid reconstruction loss decrease. In figure 5, we show how hybrid loss, SSIM, and MSE change with the number of obtained glimpses when the controlled saccade is used.

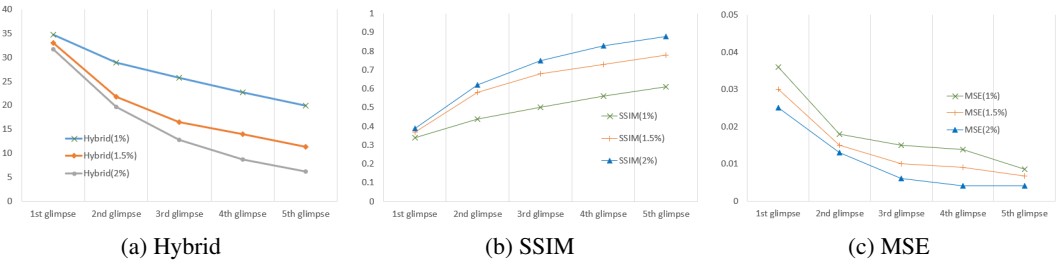

(a) Hybrid      (b) SSIM      (c) MSE

Figure 5: Hybrid Loss, SSIM, and MSE after each glance. Size of foveal region is 1/16 of the original image and the peripheral sampling proportion varies from 1% to 2%.

## A.3 RECONSTRUCTION UNDER DIFFERENT PERIPHERAL SAMPLING PROPORTION

In this section, examples under different peripheral sampling settings are provided. For all examples, the size of the foveal region is 1/16 of the original image size and the peripheral sampling proportion ranges from 1%, 2% to 5%. In each figures of Figures 6-11, there are 6 rows of images, which can be divided into three groups corresponding to the 3 peripheral sampling settings arranged in the ascending order of sampling proportion. In each group, the first row gives the sequence of input foveal-peripheral views, and the second row gives the corresponding reconstructed images. All foveal centers are selected by the saccade controller, except the one at timestep 0, which is selected randomly. The image with a green bounding box is the one that is classified correctly. Please note that the image of Monalisa does not belong to the ImageNet, therefore, it has no classification result. We can also see that some of the cases have fewer glimpses than other. This is because of the early stop mechanism. If the same classification result is recived with sufficence confidence consecutively, then the whole process will stop.

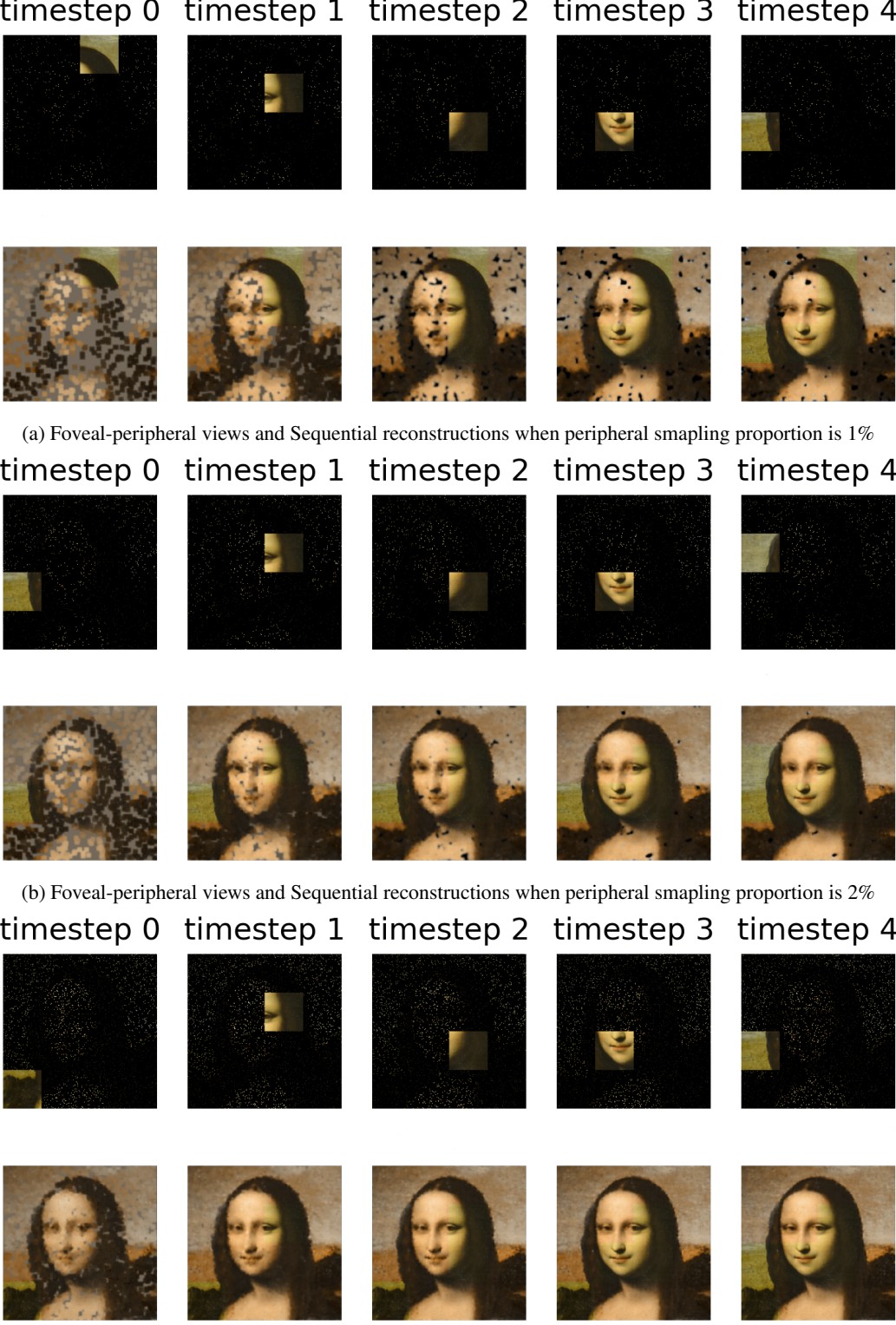

Figure 6: Monalisa reconstruction result when peripheral sampling proportion is 1%, 2% and 5% respectively.

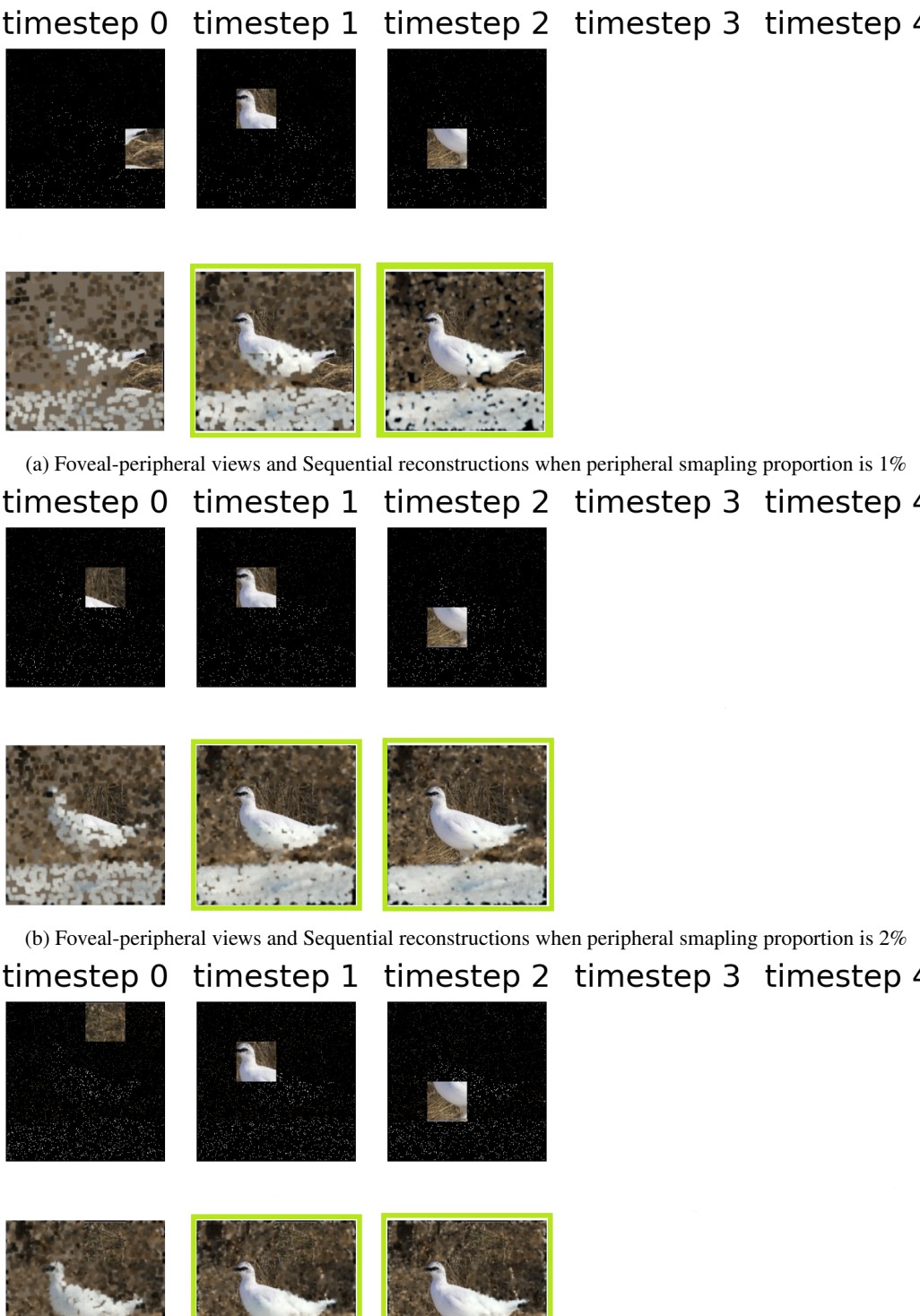

(a) Foveal-peripheral views and Sequential reconstructions when peripheral smapling proportion is 1%

(b) Foveal-peripheral views and Sequential reconstructions when peripheral smapling proportion is 2%

(c) Foveal-peripheral views and Sequential reconstructions when peripheral smapling proportion is 5%

Figure 7: ImageNet reconstruction result when peripheral sampling proportion is 1%, 2% and 5% respectively

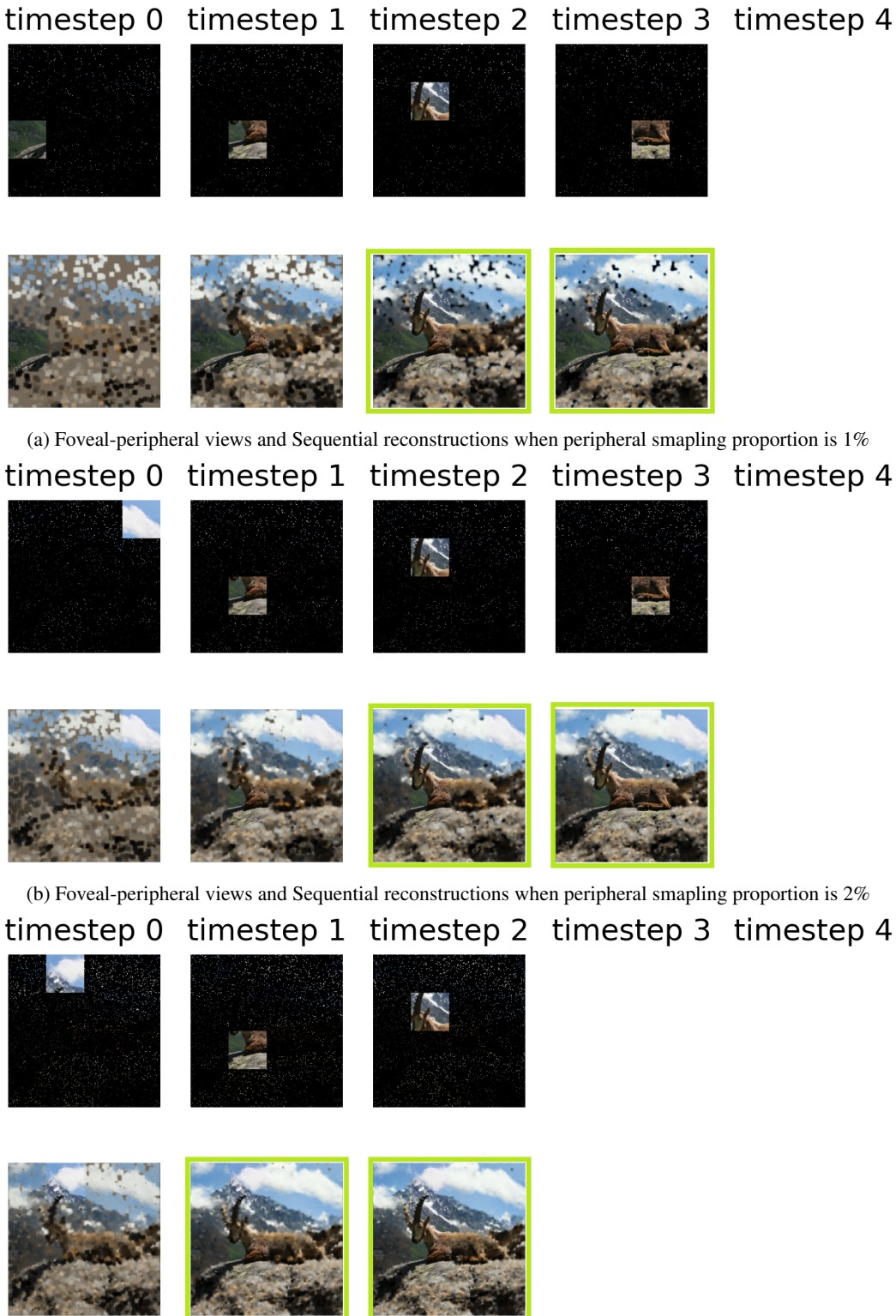

(a) Foveal-peripheral views and Sequential reconstructions when peripheral smapling proportion is 1%

(b) Foveal-peripheral views and Sequential reconstructions when peripheral smapling proportion is 2%

(c) Foveal-peripheral views and Sequential reconstructions when peripheral smapling proportion is 5%

Figure 8: ImageNet reconstruction result when peripheral sampling proportion is 1%, 2% and 5% respectively

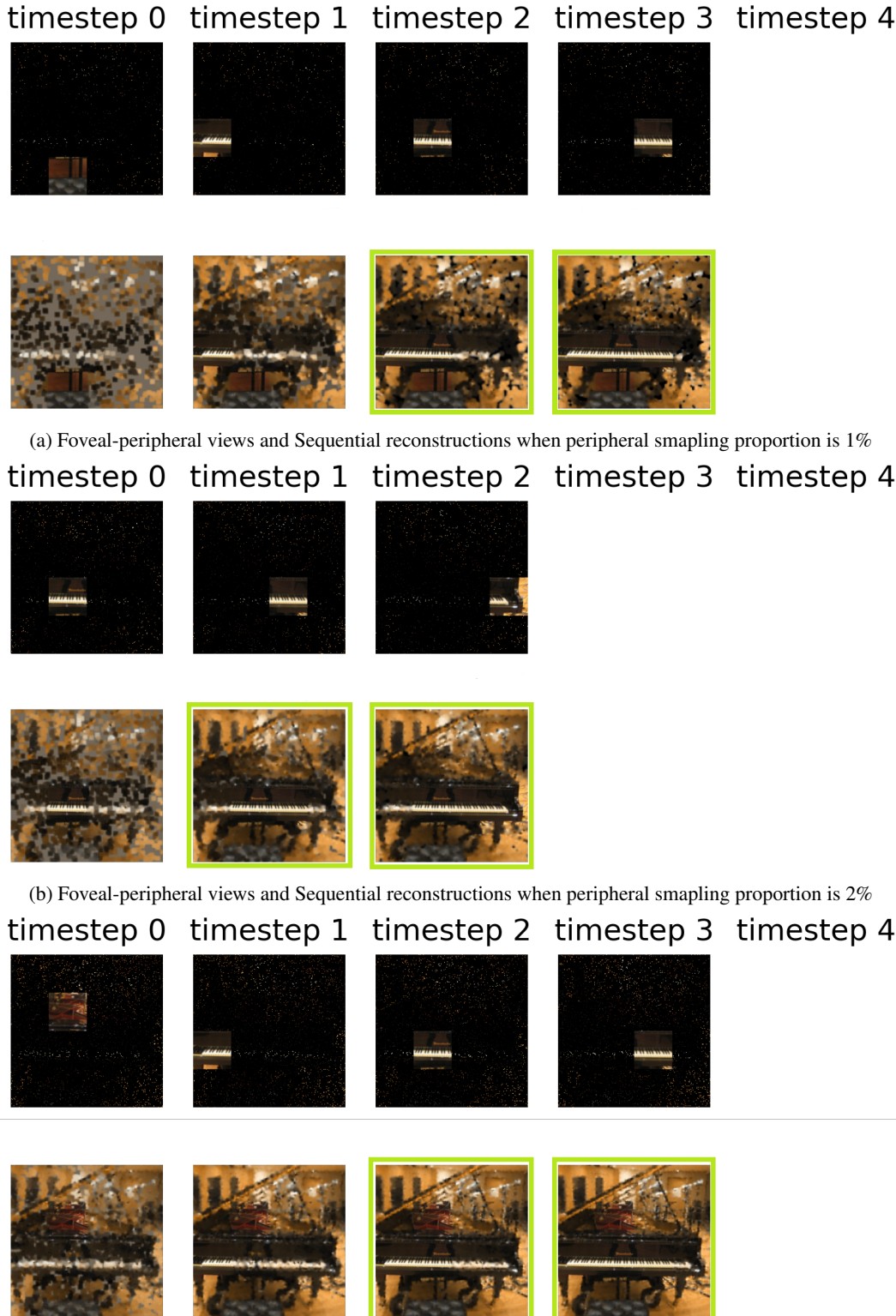

(a) Foveal-peripheral views and Sequential reconstructions when peripheral smapling proportion is 1%

(b) Foveal-peripheral views and Sequential reconstructions when peripheral smapling proportion is 2%

(c) Foveal-peripheral views and Sequential reconstructions when peripheral smapling proportion is 5%

Figure 9: ImageNet reconstruction result when peripheral sampling proportion is 1%, 2% and 5% respectively

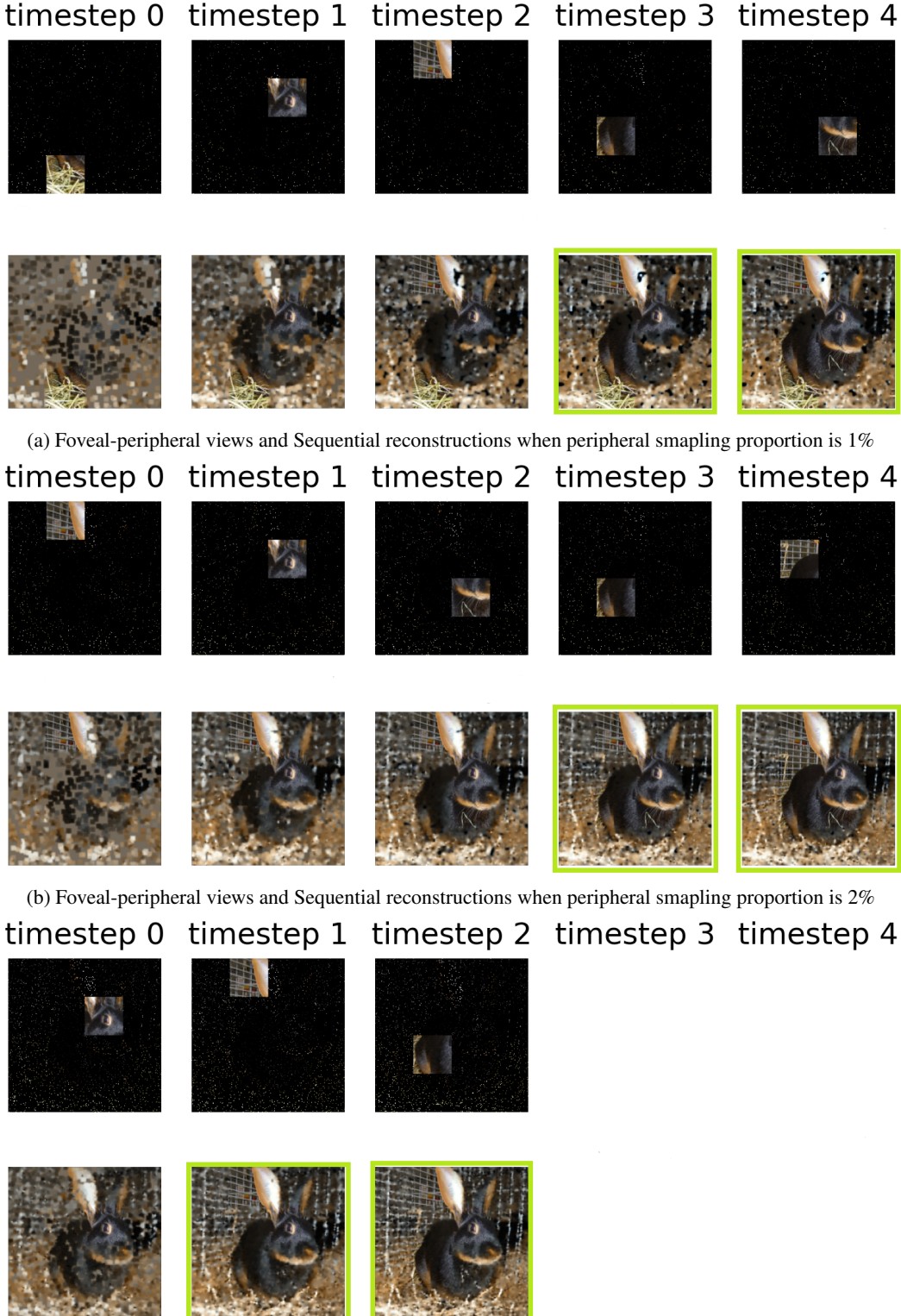

(a) Foveal-peripheral views and Sequential reconstructions when peripheral smapling proportion is 1%

(b) Foveal-peripheral views and Sequential reconstructions when peripheral smapling proportion is 2%

(c) Foveal-peripheral views and Sequential reconstructions when peripheral smapling proportion is 5%

Figure 10: ImageNet reconstruction result when peripheral sampling proportion is 1%, 2% and 5% respectively

timestep 0   timestep 1   timestep 2   timestep 3   timestep 4

(a) Foveal-peripheral views and Sequential reconstructions when peripheral smapling proportion is 1%

timestep 0   timestep 1   timestep 2   timestep 3   timestep 4

(b) Foveal-peripheral views and Sequential reconstructions when peripheral smapling proportion is 2%

timestep 0   timestep 1   timestep 2   timestep 3   timestep 4

(c) Foveal-peripheral views and Sequential reconstructions when peripheral smapling proportion is 5%

Figure 11: ImageNet reconstruction result when peripheral sampling proportion is 1%, 2% and 5% respectively

