# OpenReview forum: "Improved Efficiency Based on Learned Saccade and Continuous Scene Reconstruction From Foveated Visual Sampling"
_ICLR.cc/2024/Conference — ICLR 2024 spotlight_

### Official Review · Reviewer_7Zf9 · 2023-10-30

**Soundness:** 3 good
**Presentation:** 3 good
**Contribution:** 3 good
**Rating:** 8
**Confidence:** 3

**Summary:**

This paper presents a new algorithm for sequential foveated visual sampling of an image.

The main claims of the paper are that

- the required input pixels per frame are reduced by 90% without losing image recognition performance
- 5% higher recognition accuracy compared to existing foveal sampling models with matching pixel number input
- higher data efficiency in training

I find the algorithm to be interesting and novel, and that the second and third claims above are supported.
I am confused where to find evidence for the first claim.

Overall I think this paper is a borderline accept.

**Strengths:**

I find the method simple and useful, with interesting potential application.
It is appealing that the method seems to be suitable for existing classification models (no retraining).

**Weaknesses:**

## Major

I am confused how the image information from the sequential glimpses is passed and integrated in the predictive reconstruction model. Much more space is spent on the background to the hybrid loss function than actually making explicit how the sequential image information is used to improve reconstruction.

In addition, the abstract states "our model reduces the required input pixels by over 90% per frame while maintaining the same level of performance in image recognition as with the original images." I don't understand where to find support for this claim in the results. For example, in Figure 3, all subsampled models perform worse than the original. The data in Figure 4 are coming closest to the original; is this what is meant?

Also, please clarify whether the experiments in Figure 3 are conducted with the trained saccade control model (which one?).


## Minor

- Instead of "continuous saccades" a better terminology would be "sequential saccades" or "scanpaths". See e.g. [2, 3, 4]
- There are now known to be three types of photosensitive cells: rods, cones and intrinsically-photosensitive ganglion cells [1, 8]
- You use SSIM but the relevant paper(s) are not cited (e.g. [7]).
- Heading 3.1 "Periphrl"

## Literature

1. Do, M. T. H., & Yau, K.-W. (2010). Intrinsically Photosensitive Retinal Ganglion Cells. Physiol Rev, 90.

1. Hoppe, D., & Rothkopf, C. A. (2019). Multi-step planning of eye movements in visual search. Scientific Reports, 9(1), 144. https://doi.org/10.1038/s41598-018-37536-0

1. Kümmerer, M., & Bethge, M. (2021). State-of-the-Art in Human Scanpath Prediction (arXiv:2102.12239). arXiv. http://arxiv.org/abs/2102.12239

1. Kümmerer, M., Bethge, M., & Wallis, T. S. A. (2022). DeepGaze III: Modeling free-viewing human scanpaths with deep learning. Journal of Vision, 22(5), 7. https://doi.org/10.1167/jov.22.5.7

1. Rosenholtz, R. (2016). Capabilities and Limitations of Peripheral Vision. Annual Review of Vision Science, 2(1), 437–457. https://doi.org/10.1146/annurev-vision-082114-035733

1. Watson, A. B. (2014). A formula for human retinal ganglion cell receptive field density as a function of visual field location. Journal of Vision, 14(7), 15. https://doi.org/10.1167/14.7.15

1. Wang, Z., Simoncelli, E. P., & Bovik, A. C. (2003). Multiscale structural similarity for image quality assessment. The Thirty-Seventh Asilomar Conference on Signals, Systems & Computers, 2003, 1398–1402. https://doi.org/10.1109/ACSSC.2003.1292216

1. Zele, A. J., Feigl, B., Adhikari, P., Maynard, M. L., & Cao, D. (2018). Melanopsin photoreception contributes to human visual detection, temporal and colour processing. Scientific Reports, 8(1), 3842. https://doi.org/10.1038/s41598-018-22197-w

**Questions:**

- I would like to see how the hybrid reconstruction loss changes over timestep, and not just classification accuracy.
- The sampling of the periphery of individual pixels with small probability is not very like human vision. Effectively this is providing low pass information. Have the authors considered how the sampling density could be approximated more plausibly (e.g. [6])?
- Have the authors considered comparing scanpath strategies learned in this model to human scanpaths (e.g. [3, 4])?

---

> ### Author Response · Authors · 2023-11-18
> **Authors' replies to Reviewer 7Zf9(1/2)**
>
> Thank you for the valuable feedback. A revision paper has been uploaded. Below is our response. The original comments are copied followed by our answers.
>
> >Q1: I would like to see how the hybrid reconstruction loss changes over timestep….
>
> A: Three plots were added in Appendix A.2 to illustrate how hybrid loss, SSIM, and MSE changes with the number of glimpses on the ImageNet dataset. Data in plots were generated from three different foveal-peripheral settings with controlled saccade. The size of the fovea region is fixed to 1/16 of the original image, and the peripheral sampling proportion varies from 1% to 2%. As we can see, the hybrid loss and MSE reduces and SSIM increases with the number of glimpses.
>
> >Q2: The sampling of the periphery of individual pixels with small probability is not very like human vision. Effectively this is providing low pass information. Have the authors considered how the sampling density could be approximated more plausibly?
>
> A: We agree with you that our sampling of peripheral vision is not very biomimetic. We chose this simple technique for its low-cost, hardware friendliness, and also because it results in a sparse input, which is our goal. Some existing works transform a conventional high quality image into foveated images using gaussian blur or texture-based transformation[1]. However, these foveated images are not sparse. Every pixel in the image still has a non-zero value, hence they still have to be transmitted and processed. Some other works sample periphery using more biologically plausible techniques, such as log-rectilinear transformation[2][3]. However, the cost for a camera to support log-rectilinear sampling will be too expensive. Nonetheless, it is our future plan to compare different sampling techniques, to understand the choice of biological systems and also to find tradeoffs between cost and performance.
>
>
> [1]Deza, A., et. al.  “Towards metamerism via foveated style transfer.”
>
> [2]Martínez, J., et. al. “A new foveal cartesian geometry approach used for object tracking.”
>
> [3]D. Li, et. al., "A Log-Rectilinear Transformation for Foveated 360-degree Video Streaming"
>
> >Q3:Have the authors considered comparing scanpath strategies learned in this model to human scanpaths?
>
> A: This is a very interesting idea and we have included this as part of our future works in the revision paper. We need to point out that our saccade model is trained specifically for a targeted functional backend. Unless the scanpath is recorded when a human subject is performing the same visual task, such as image classification, we should not expect these two to be the same. However, it is still interesting to find out how they differ from each other.

---

> ### Author Response · Authors · 2023-11-18
> **Authors' replies to Reviewer 7Zf9(2/2)**
>
> >Major W1:I am confused how the image information from the sequential glimpses is passed and integrated in the predictive reconstruction model.
>
> A: Thanks for pointing that out. We added Appendix A.1 in the revised paper to provide more explanations about the predictive reconstruction model. The model has three-layers of ConvLSTM and one convolutional layer. It predicts the reconstructed image using the current foveal-peripheral view sampled at current glimpse, and the history information that is carried in the hidden states.  The first two layers of the ConvLSTM gradually increase the depth of feature maps, while the last layer compresses all the generated feature maps into the reconstructed image.
>
> >Major W2: … In Figure 3, all subsampled models perform worse than the original. The data in Figure 4 are coming closest to the original; is this what is meant?
>
> A: We apologize for not making it clear. You are correct, when we say “our model reduces the required input pixels by over 90% per frame while maintaining the same level of performance”, we mean the results in Figure 4. Figure 3 shows the results only for image reconstruction without trained saccade, while Figure 4 shows the results with trained saccade. Because we will sample 1/16 image at the foveal center and sample 2% pixels as peripheral vision, the number of pixels sampled at each time step is 8.25% of the original image, which corresponds to 90% of pixel reduction per frame.
>
> >Major W3: Also, please clarify whether the experiments in Figure 3 are conducted with the trained saccade control model (which one?).
>
> A: Experiments in Figure 3 are conducted without trained saccade. We have modified the figure caption and added this information.
>
> >Minor W1:” Instead of "continuous saccades" a better terminology would be "sequential saccades" or "scanpaths". See e.g. [2, 3, 4]
>
> A: We appreciate your suggestion, and we have changed the terms in our paper as suggested.
>
> >Minor W2: …There are now known to be three types of photosensitive cells …
>
> A: Thanks for pointing this out. We have rewritten the relevant sentences in section 2.1 to emphasize retinal image capture and formation. The extent to which ipRGCs participate is somewhat controversial and not central to the focus of this paper.
>
> >Minor W3: SSIM paper is not cited.
>
> A: Thank you for pointing it out. The reference is added in the revision paper.
>
> >Minor W4: Heading 3.1 "Periphrl"
>
> A: Thank you for pointing this out. We have corrected this typo.

---

> > ### Comment · Reviewer_7Zf9 · 2023-11-22
> > **Review score unchanged**
> >
> > Thanks to the authors for their detailed responses. My review score remains unchanged.

---

> ### Author Response · Authors · 2023-11-22
> **Gentle reminder Regarding Rebuttal: 8429**
>
> Dear Reviewer 7Zf9,
>
> We do appreciate your positive opinion on our work, and it means a lot to us.
>
> With the rebuttal deadline approaching, it will be great if we can hear your thoughts on our response and revisions. Have we adequately addressed your concerns? Your feedback is very important to us, and we are more than happy to further improving our paper based on your insights.
>
> Thank you for your time.
>
> Best regards,
>
> Authors

---

> ### Author Response · Authors · 2023-11-22
> **To Reviewer 7Zf9**
>
> Dear Reviewer,
>
> We sincerely appreciate the time and effort you spent on our work. Thank you!
>
> Best regards,
>
> Authors

---

### Official Review · Reviewer_rEUv · 2023-10-30

**Soundness:** 2 fair
**Presentation:** 3 good
**Contribution:** 3 good
**Rating:** 6
**Confidence:** 4

**Summary:**

This paper aims to reconstruct the original image from multiple subsampled views, using reinforcement learning and neural network models for scan control and image reconstruction, respectively. The paper conducts numerous experiments to demonstrate that the proposed algorithm can maintain detection task accuracy, reasonable saccade control, and high reconstruction quality under high data efficiency. However, the motivation for the work is not well-founded, and there are possible improvements in the experiments.

**Strengths:**

1. The task addressed in the paper is novel, as it is the first in the industry to reconstruct an image from continuous central foveal subsampled images. Other methods focus on single-sample images and proceed directly to downstream tasks without reconstructing the original image, making this work unique.
2. The methods used are innovative, employing an actor-critic model for saccade control, which can achieve near-original image classification accuracy in just five scans.
3. The writing style of the paper is easy to understand, especially in describing the proposed methods.

**Weaknesses:**

1. While the task is novel, it lacks a convincing real-world application, as it simulates the process of multiple eye samplings without addressing practical problems.
2. The experimental comparisons are not entirely fair. The uniform control group uses an 8% sampling probability, while the 1/16+2% group differs by 0.25%, indicating an unequal amount of information that might affect performance.
3. Using classification model metrics to assess the quality of reconstruction is questionable, as classification tasks do not focus on texture details. If this method was to downsample the original image with the same number of sampled pixels, how much better is the method in terms of performance compared to this?

**Questions:**

see weaknesses

---

> ### Author Response · Authors · 2023-11-18
> **Authors' reply to Reviewer rEUv**
>
> We appreciate the reviewer for the insightful questions. A revision paper has been uploaded. Below is our response. The original comments are copied followed by our answers.
>
> >Q1: … lacks a convincing real-world application…
>
> A: While our motivation was discussed in the abstract section, we acknowledge the oversight in not highlighting it extensively later in the paper, constrained by space limitations. Our goal is to build a pixel efficient visual system. Our approach decreases the volume of incoming pixels (from the camera) while maintaining the performance of backend visual applications and simplifies camera design and cost, increases frame rate, and reduces the energy and latency for data transmission between the camera and processor. Furthermore, previous research indicated that the texture-based foveal representation also offers higher adversarial and occlusion robustness. The integrated saccade controller and predictive reconstruction model allow us to achieve an accuracy comparable to the original image without retraining or fine tuning the functional backend, such as an image classifier.
>
> >Q2: …The uniform control group uses an 8% sampling probability, while the 1/16+2% group differs by 0.25%....
>
> A: Thank you so much for pointing out this discrepancy. We have increased the sampling probability of the uniform control group to 8.25% and updated Figure 3 in the revised paper. We note that the new results are very similar to the old ones.
>
> >Q3a: Using classification model metrics to assess the quality of reconstruction is questionable ….
>
> A: Thanks for raising this concern. The goal of our work is not to just reconstruct the image, but to use fewer camera pixels to achieve a good performance in the functional backend. Image classification is just an example of backend application. Our framework can support other backend applications, such as object detection, object tracking, etc. We train the saccade model to maximize the performance of the targeted backend application.
>
> Notably, using a saccade model trained with the feedback from an image classifier, the reconstructed images (Table 2, row 3~8) have a lower structural similarity (SSIM) to the original image, but a higher accuracy in classification. However, using a saccade model trained directly to maximize the SSIM of the reconstruction (row 2), the reconstructed images have a higher SSIM but a lower classification accuracy.  In other words, the saccade model should be trained using the performance of  targeted backend applications as a feedback.
>
> >Q3b: If this method was to downsample the original image with the same number of sampled pixels, how much better is the method in terms of performance compared to this?
>
> A: In Figure 3, the data points “uniform, no conn” and “uniform” give the performance of this case.  The “uniform, no conn” down samples images to 8.25% of their original size without reconstruction; while “uniform” reconstruct the images after downsampling. As we can see, the classification accuracy of downsampled images without reconstruction is less than 10%. The accuracy increases to 75.3% and 57% for GTSRB and ImageNet dataset respectively after reconstruction. However, it is still lower than the accuracy (88.9% and 62.6%) of the reconstructed foveal-peripheral view that has the same number of pixels (the red data points in Figure 3). With a trained saccade model, the classification accuracy increases to 91.8% and 73.7% for GTSRB and ImageNet respectively.

---

> ### Author Response · Authors · 2023-11-21
> **Gentle reminder Regarding Rebuttal: 8429**
>
> Dear Reviewer rEUv,
>
> With the rebuttal deadline approaching, it will be great if we can hear your thoughts on our response and revisions. Have we adequately addressed your concerns? Your feedback is crucial to us, and we're committed to further improving our paper based on your insights.
>
> Thank you for your time.
>
> Best regards,
>
> Authors

---

> > ### Comment · Reviewer_rEUv · 2023-11-22
> >
> > Thank you for addressing my concerns.  I'm pleased to confirm that the issues have been resolved.

---

> ### Author Response · Authors · 2023-11-22
> **To Reviewer rEUv**
>
> Dear Reviewer,
>
> We sincerely appreciate the time and effort you spent on our work. Thank you!
>
> Best regards,
>
> Authors

---

### Official Review · Reviewer_cXLY · 2023-10-31

**Soundness:** 3 good
**Presentation:** 2 fair
**Contribution:** 2 fair
**Rating:** 6
**Confidence:** 2

**Summary:**

The authors present an innovative solution for image classification and detection that addresses the trade-off between image quality and computational efficiency. They introduce an active scene reconstruction architecture that leverages foveal and peripheral views, along with a reinforcement learning-based saccade mechanism, reducing input pixels by over 90% per frame while maintaining image recognition performance.

**Strengths:**

- The paper introduces an innovative concept inspired by the human visual system, combining foveal and peripheral views with a saccade mechanism in image reconstruction. This approach has potential applications in various fields.
- A 90% reduction in required input pixels per frame has practical implications for real-time image processing
- paper is easy to read

**Weaknesses:**

- Although the paper addresses the trade-off between image quality and computational efficiency, it would be valuable to provide insights into the computational overhead of implementing the proposed model, particularly in terms of hardware and energy requirements.
- The paper totally fails to mention a whole branch of literature in saccade modeling. See for example [1], [2], or  [3]. In particular, [2] also uses reconstruction as a guiding task. It seems true that none of the mentioned approaches focused on performance in terms of image reconstruction, but I think it is relevant to at least position the current contribution compared to those. I imagine, some of these saccade models could potentially be used in the same framework proposed by the authors here.

[1] Wloka, C., Kotseruba, I., & Tsotsos, J. K. (2018). Active fixation control to predict saccade sequences. In Proceedings of the IEEE Conference on Computer Vision and Pattern Recognition (pp. 3184-3193).
[2] Schwinn, L., Precup, D., Eskofier, B., & Zanca, D. (2022). Behind the Machine’s Gaze: Neural Networks with Biologically-inspired Constraints Exhibit Human-like Visual Attention. Transactions on Machine Learning Research.
[3] Assens, M., Giro-i-Nieto, X., McGuinness, K., & O'Connor, N. E. (2018). PathGAN: Visual scanpath prediction with generative adversarial networks. In Proceedings of the European Conference on Computer Vision (ECCV) Workshops (pp. 0-0).

**Questions:**

- Can you provide more details about the computation overhead of your model and possible complications in real world applications?
- Can you better frame your contribution, and compare it to the literature in saccade modeling?

---

> ### Author Response · Authors · 2023-11-18
> **Authors' replies to Reviewer cXLY(1/2)**
>
> We appreciate the valuable feedback from the reviewer. A revision paper has been uploaded. Below is our response. The original comments are copied followed by our answers.
>
> >Q1:  Can you provide more details about the computation overhead of your model and possible complications in real world applications?
>
> A: . By leveraging foveal-peripheral representation, we can reduce the amount of required pixels by 60-70%. (Please see Table 2 for the pixel usage and test accuracy). This reduction can help to simplify the camera design, increase the frame rate, and reduce the energy for data transmission between camera and processor. The computation overhead is the extra step of image reconstruction and saccade control before the backend applications such as image classifying. The reconstruction model and saccade control model totally have 24.5M trainable parameters, which is about half of the size of ResNet101. In terms of computation complexity, in each glimpse, the reconstruction and saccade model totally have 4.3G floating point operations (FLOPs) (0.7G FLOPs for reconstruction and 3.6G FLOPs for saccade control). We did not include the computation complexity of the critic network of the saccade controller, as it is not needed in the execution time. As a reference, ResNet101 has 15.7G FLOPs per image.
>
> During real-world applications, the foveal-peripheral view will first be sent to the reconstruction model. The hidden state of the reconstruction model will be sent to the saccade controller for saccade action and the reconstructed image will be sent to the backend model for further classification. A potential complication in real world application is that multiple glimpses (3.5 ~ 7 glimpses on average) may be needed for the backend model to reach a performance comparable to the original image. (Please note that the pixel usage reported in Table 1 is the total number of pixels used by all glimpses.) However, in a real world environment, the scene change is usually slower than the camera frame rate and the forward propagation time of the reconstruction model and saccade controller. The latency of multiple saccade movements can be hidden.

---

> ### Author Response · Authors · 2023-11-18
> **Authors' replies to Reviewer cXLY(2/2)**
>
> >Q2: Can you better frame your contribution, and compare it to the literature in saccade modeling?
> A: Thank you for pointing out prior works ( [1][2][3]) in saccade model and scanpath generation. We have cited them in the revised paper.
>
> Among these papers, reference [1] splits images into central path and peripheral path for saliency extraction and combines them to form a conspicuity map. The fovea center is picked based on the bottom up saliency map. This, to some degree, is similar to “Lukanov et al. (2021)” (cited in section 2.2 of our paper). The limitation of saliency based saccade is that it ignores requirements from high level applications. “Lukanov et al. (2021)” mitigated this limitation by combining the saliency information with a top-down feedback, which is the activation map from a high level classifier. This again has a problem. At the beginning of the visual process, with very few glimpses, the classification is likely to be wrong, which will lead to a wrong position for the next saccade movement. Furthermore, they trained the saccade model and the classifier together. The mutual dependency on each other makes the training difficult. While our approach selects the next saccade movement based on the hidden state of the reconstruction model, which has been pre-trained. The reward to train the saccade policy comes from an existing classifier working on reconstructed images. By training these models separately, our flow is simpler and achieves better results.
>
> As explained above, image reconstruction is not our final goal, it is an intermediate step to facilitate the training of the saccade controller using the feedback from an existing upper level application model. Although reference [2] also considers image reconstruction, they perform it as a final goal (i.e., upper level application.) Furthermore, the input image of [2] is a foveation-blur representation generated by a Gaussian filter. Technically speaking, foveation-blur representation of the image is not sparse. Every pixel in the image still has a non-zero value, hence they still have to be transmitted and processed. We did not use the foveation-blur representation in our work because our goal is to reduce the amount of input data from the camera.
>
> Reference [3] generates a visual scanpath for a given image using generative adversarial networks (GAN). They do not consider the foveation effect. The GAN needs the original image with full resolution to generate the scanpath. Furthermore, to train this GAN, recorded human eye movement is needed as labeled data. Therefore, the training process will be very expensive.
>
> [1] Wloka, C., Kotseruba, I., & Tsotsos, J. K. (2018). Active fixation control to predict saccade sequences. In Proceedings of the IEEE Conference on Computer Vision and Pattern Recognition (pp. 3184-3193).
>
> [2] Schwinn, L., Precup, D., Eskofier, B., & Zanca, D. (2022). Behind the Machine’s Gaze: Neural Networks with Biologically-inspired Constraints Exhibit Human-like Visual Attention. Transactions on Machine Learning Research.
>
> [3] Assens, M., Giro-i-Nieto, X., McGuinness, K., & O'Connor, N. E. (2018). PathGAN: Visual scanpath prediction with generative adversarial networks. In Proceedings of the European Conference on Computer Vision (ECCV) Workshops (pp. 0-0).

---

> ### Author Response · Authors · 2023-11-21
> **Gentle reminder Regarding Rebuttal: 8429**
>
> Dear Reviewer cXLY,
>
> As the rebuttal deadline nears, it would be great if we can receive your updated opinion on our response and revisions. Have we sufficiently addressed your concerns? Your insights are pivotal to us, and we're dedicated to refining our paper based on your valuable feedback.
>
> Appreciate your time.
>
> Best regards,
>
> Authors

---

> > ### Comment · Reviewer_cXLY · 2023-11-22
> > **response to authors**
> >
> > Thanks for your comments, I am satisfied with them and I increased my score accordingly.

---

> ### Author Response · Authors · 2023-11-22
> **To Reviewer cXLY**
>
> Dear Reviewer,
>
> We sincerely appreciate the time and effort you spent on our work. Thank you!
>
> Best regards,
>
> Authors

---

### Official Review · Reviewer_5v5k · 2023-11-03

**Soundness:** 2 fair
**Presentation:** 2 fair
**Contribution:** 3 good
**Rating:** 8
**Confidence:** 5

**Summary:**

This paper presents a novel application of spatially-varying computation (foveation) coupled with eye-movements towards the goal of image reconstruction. The authors introduce an active sensing model that takes into account all of the image information in a spatially varying way and continually updates the visual stimulus until it is near perfectly reconstructed. Authors introduce a novel loss function and show small toy experiments that prove their claims.

**Strengths:**

- This paper presents a novel application of foveal-peripheral vision tailored towards image reconstruction.
- The paper has shown and presented a set of experiments that seem to support their claimed contribution
- The paper has references many other works in perceptual psychology and neuroscience -- though many more of these papers are missing, and the field has moved forward quite a lot (see Weaknesses below), thus potentially impacting the novelty of the paper.

**Weaknesses:**

I think the main weakness this paper has is I am confused on how the system is trained to do reconstruction. Is it doing the reconstruction from the same image and "testing on the training set"? Otherwise, I am surprised the first auto-completion of the image is surprisingly quite well without any prior knowledge of the underlying geometry of the visual stimulus. If indeed it is testing on the training set, what would be the contribution/application of such system? A compression engine that works better than JPEG, or would the contribution here really be more of an intellectual one of saying that reconstruction through foveation is indeed possible.

-------
There are a set of missing papers that the authors should add and/or discuss in this work. While none of these papers directly attack the problem of using foveation as a tool for reconstruction, many of such works discuss the complimentary theory of foveation having a representational goal in addition to purely optimizing for metabolic cost (and thus limiting the impact of the authors through this paper)

Key Missing Critical References:
- Deza & Konkle. ArXiv, 2021. Emergent Properties of Foveated Perceptual Systems.
- Wang & Cottrell. Journal of Vision, 2017. Central and peripheral vision for scene recognition: A neurocomputational modeling exploration.
- Cheung, Weiss & Olshausen. ICLR 2017. Emergence of foveal image sampling from learning to attend in visual scenes

Secondary, but also important References:
- Gant, Banburski & Deza. SVRHM, 2022. Evaluating the adversarial robustness of a foveated texture transform module in a CNN.
- Reddy, Banburski, Pant & Poggio. NeurIPS 2020. Biologically inspired mechanisms for adversarial robustness
- Wang, Mayo, Deza, Barbu & Conwell. SVRHM, 2021. On the use of Cortical Magnification and Saccades as Biological Proxies for Data Augmentation
- Harrington & Deza. ICLR, 2022. Finding Biological Plausibility for Adversarially Robust Features via Metameric Tasks

In addition the original SSIM paper:
- Wang, Bovik, Sheik & Simoncelli. IEEE TIP, 2004. Image quality assessment: from error visibility to structural similarity (SSIM).

and Foveation paper that introduce the idea of texture-based computation in the periphery:
- Freeman & Simoncelli. Nature Neuroscience, 2011. Metamers of the Ventral Stream.

**Questions:**

I am open to changing my mind about this paper. There are a lot of missing papers, but the idea seems interesting. I am fan of papers that explore non-intuitive applications or theories of foveation but I am still not there yet to give this paper a clear accept.

I'm also struggling to know what is $t_0$? Is it a blank image? Is it a corrupted image? Is it only a fraction/glimpse of an image?

---

> ### Author Response · Authors · 2023-11-18
> **Authors' replies to Reviewer 5v5k (1/2)**
>
> We thank the reviewer for the valuable feedback. A revision paper has been uploaded. Below is our response to your questions and concerns. The original comments are copied followed by our answers.
>
> >W1: …the first auto-completion of the image is surprisingly quite well. Is it doing the reconstruction from the same image and "testing on the training set"?
>
> A:  Thank you for raising this question. The reconstruction model is trained and tested on traning set and testing set respectively. It is not “testing on the training set”. Although the model has no prior knowledge of the geometry of the object, it does receive inputs from the peripheral region. The peripheral sampling portion is 2% for the Monalisa image shown in Figure 2 and 5% for the example images in the Appendix in the original paper. Although the peripheral input is too sparse for humans to recognize any geometry information, the reconstruction model is capable of roughly completing the image by filling up the gaps. In the revised paper, we added more examples to the Appendix A.3 to showcase the reconstruction. In those examples, the peripheral sampling portion varies from 1% to 2% to 5%. As we can see, even with a 2% sampling portion, the model can already roughly detect the shape of the object after the 1st glimpse. The examples also include the reconstruction of Monalisa, which is not part of the imageNet data set, as evidence that our testing data is different from the training data.
>
> >W2: There are a set of missing papers … and thus limiting the impact of the authors through this paper.
>
> A: Thank you so much for pointing out these references. We have cited them in our revised version. Due to the space limit, we are not able to discuss and compare to each one of them in the paper. We need to point out that, in general, these works have a different focus compared to ours.
>
> Thank you for the  references [1][5][6][7][8]. Indeed they provide useful background in showing that the visual crowding effect in the texture-based fovea representation of the primate retina provides adversarial robustness and efficiency. This is another reason why it is important to study foveal-peripheral visual systems. We have mentioned this in section 1 in the revised paper. However, the goal of these papers is to demonstrate a property of foveated images, while our goal is to build a processing pipeline and saccade mechanism for the foveated images.
>
> Reference [2] shares some similarity with (Jaramillo-Avila & Anderson, 2019) cited in section 2.2 of our paper, in the sense that both works trained a backend model for a specific application with foveal-peripheral input. As we mentioned in the paper, the issue with training a new backend model for the foveal-peripheral input is the need for a large amount of labeled data in order to reach the state-of-art performance. By using unsupervised training to reconstruct the image, we can leverage the existing backend model without re-train them.
>
> Reference [3] is somewhat similar to (Chen, 2021) that we cited in section 2.2. Both papers take several decision steps to find the location of the foveal center and, for each image, only one foveal center is identified. Furthermore, the goal of [3] is to locate the target as quickly as possible.
>
> [1]Deza & Konkle. ArXiv, 2021. Emergent Properties of Foveated Perceptual Systems.
>
> [2]Wang & Cottrell. Journal of Vision, 2017. Central and peripheral vision for scene recognition: A neurocomputational modeling exploration.
>
> [3]Cheung, Weiss & Olshausen. ICLR 2017. Emergence of foveal image sampling from learning to attend in visual scenes
>
> [4]Wang, Bovik, Sheik & Simoncelli. IEEE TIP, 2004. Image quality assessment: from error visibility to structural similarity (SSIM).
>  interesting, and include
>
> [5]Gant, Banburski & Deza. SVRHM, 2022. Evaluating the adversarial robustness of a foveated texture transform module in a CNN.
>
> [6]Reddy, Banburski, Pant & Poggio. NeurIPS 2020. Biologically inspired mechanisms for adversarial robustness
>
> [7]Wang, Mayo, Deza, Barbu & Conwell. SVRHM, 2021. On the use of Cortical Magnification and Saccades as Biological Proxies for Data Augmentation
>
> [8]Harrington & Deza. ICLR, 2022. Finding Biological Plausibility for Adversarially Robust Features via Metameric Tasks

---

> ### Author Response · Authors · 2023-11-18
> **Authors' replies to Reviewer 5v5k (2/2)**
>
> >Q1: I'm also struggling to know what is t0? Is it a blank image? Is it a corrupted image? Is it only a fraction/glimpse of an image?
>
> A: We couldn’t find where variable t0 is mentioned in the paper except in equation (4), where we tried to sum the gradient from t=0 to T-1. We must apologize that this is a typo in the equation, the summation should start from t=1 to T as the following:
>
> \begin{equation}
>     \bigtriangledown_{\theta_p} J(\theta_p) =E[\sum_{t=1}^{T}\bigtriangledown_{\theta_p}log{\pi\theta_p}(a_{t}|s_{1:t})A(s_{1:t},a_t)],
> \end{equation}
>
> If the question is about the first input at time t=0, then the first input is the foveal-peripheral view of the image with a randomly chosen fovea center. We have added this information in the revised paper.

---

> ### Author Response · Authors · 2023-11-21
> **Gentle reminder Regarding Rebuttal:8429**
>
> Dear Reviewer 5v5k,
>
>
> With the rebuttal deadline approaching, we're eager to hear your thoughts on our response and revisions. Have we adequately addressed your concerns? Your feedback is crucial to us, and we're committed to further improving our paper based on your insights.
>
>
> Thank you for your time.
>
> Best regards,
>
> Authors

---

> > ### Comment · Reviewer_5v5k · 2023-11-22
> > **Increased my Score , Concerns addressed**
> >
> > Dear Authors,
> >
> > I have seen the revised version of the paper, and I am definitely more happy with it. Figure 2 is more clear and Figure 3 looks better as well. I think this paper would make a good poster for ICLR.
> >
> > There is also a small typo in the added references: The Harrington & Deza SVRHM 2022 paper is actually also a ICLR 2022 (spotlight) paper and would be best to cite the ICLR version instead. I believe this is the link: https://openreview.net/forum?id=yeP_zx9vqNm
> >
> > The extra experiments for controlling for No Saccades raised by other reviewers are also important.
> >
> > On the side of reconstruction: this paper (DeepFovea) perhaps may also be important that you are welcome to cite if relevant: https://github.com/facebookresearch/DeepFovea (Of course they are missing the RL framework that the Authors have integrated, but all-in-all, may be worth citing as well!)

---

> ### Author Response · Authors · 2023-11-22
> **To Reviewer 5v5k**
>
> Dear Reviewer,
>
> Thanks for your advices! We will modify our paper accordingly.
>
> We sincerely appreciate the time and effort you spent reviewing our paper.
>
> Best regards,
>
> Authors

---

> ### Author Response · Authors · 2023-11-22
> **Revision updated**
>
> Dear reviewer,
>
> A revision has been uploaded. We cited "Deep fovea" in introduction section right before sentence "The accuracy degradation can partly be mitigated...". We also corrected the added reference to ICLR version as you suggested.
>
> Thank you for the valuable advices.
>
> Authors

---

### Author Response · Authors · 2023-11-20
**To all reviewers**

Dear Respected Reviewers,

We sincerely appreciate the time and effort you spent reviewing our paper. With the valuable feedback received, we have improved our paper and submitted the revision to OpenReview.

Below is a brief overview of the major modifications made:

* Predictive Reconstruction model Details: Additional information regarding the predictive reconstruction models is added as  in Appendix A.1, offering a more comprehensive introduction of our approach.(In response to reviewer 7Zf9 and 5v5k)

* Plots for Analysis: Three plots have been added to Appendix A.2. They illustrate the dynamic changes in hybrid loss, SSIM, and MSE over different timesteps. (In response to reviewer 7Zf9 and rEUv)

* Reconstruction Results: Appendix A.3 now includes more examples of reconstruction, specifically under various peripheral sampling proportions. This section strives to present readers with a straightforward understanding of the model's performance.(In response to reviewer 5v5k)

* Missing Citations: Your suggestions regarding missing references have been thoroughly addressed. In Section 1 and Section 2.2, we have discussed and cited the missing papers mentioned by the reviewers.(In response to reviewer cXLY, 5v5k and 7Zf9)

The changes in the paper are highlighted using red text. Once again, we genuinely appreciate your dedication to the review process and look forward to your continued insights.

Best Regards,

Authors.

---

### Meta-Review · Program_Chairs · 2023-12-05

**Metareview:**

This work proposes a reinforcement-learning based image sampling approach, based on foveal-peripheral vision, for efficient image recognition and scene reconstruction. Reviewers generally felt the approach was novel, the paper well-written, and the experimental analysis convincing. After improvements during the discussion period, the reviewers unanimously recommend acceptance. The AC sees no reason to override this recommendation.

**Justification For Why Not Higher Score:**

Reviewers raised issues with missing details of the method, as well as several missing references, as well as comparisons to non foveated systems.

**Justification For Why Not Lower Score:**

The paper is well-written, novel, and has strong results and analysis.

---

### Decision · Program_Chairs · 2024-01-16

Accept (spotlight)